# OmniVL: One Foundation Model for Image-Language and Video-Language Tasks

**Junke Wang**[1,2], **Dongdong Chen**[3], **Zuxuan Wu**[1,2†], **Chong Luo**[4], **Luowei Zhou**[3⋆],
**Yucheng Zhao**[4], **Yujia Xie**[3], **Ce Liu**[3], **Yu-Gang Jiang**[1,2†], **Lu Yuan**[3]

[1]Shanghai Key Lab of Intell. Info. Processing, School of CS, Fudan University
[2]Shanghai Collaborative Innovation Center on Intelligent Visual Computing
[3]Microsoft Cloud + AI, [4]Microsoft Research Asia

## Abstract

This paper presents OmniVL, a new foundation model to support both image-language and video-language tasks using one universal architecture. It adopts a unified transformer-based visual encoder for both image and video inputs, and thus can perform joint image-language and video-language pretraining. We demonstrate, for the first time, such a paradigm benefits both image and video tasks, as opposed to the conventional one-directional transfer (*e.g.*, use image-language to help video-language). To this end, we propose a *decoupled* joint pretraining of image-language and video-language to effectively decompose the vision-language modeling into spatial and temporal dimensions and obtain performance boost on both image and video tasks. Moreover, we introduce a novel unified vision-language contrastive (UniVLC) loss to leverage image-text, video-text, image-label (*e.g.*, image classification), video-label (*e.g.*, video action recognition) data together, so that both supervised and noisily supervised pretraining data are utilized as much as possible. Without incurring extra task-specific adaptors, OmniVL can simultaneously support visual only tasks (*e.g.*, image classification, video action recognition), cross-modal alignment tasks (*e.g.*, image/video-text retrieval), and multi-modal understanding and generation tasks (*e.g.*, image/video question answering, captioning). We evaluate OmniVL on a wide range of downstream tasks and achieve state-of-the-art or competitive results with similar model size and data scale.

## 1 Introduction

Vision-language pretraining has been demonstrated to be a promising direction for building foundation models that can support a broad range of downstream AI tasks. By pretraining on web-scale noisy image-text data, the pioneering works [53, 29, 71] suggest a unified model can be equipped with unprecedented capabilities (*e.g.*, zero-shot classification) and achieve outstanding performance on various tasks, thus significantly reducing the cost of designing task-specific models. Following this thread, some works [55, 60, 3, 37, 79] are further proposed to support more tasks. There are also some efforts [23, 72] studying video-language pretraining to solve video-related multi-modal tasks.

In this paper, we take a step forward and aim to design an omni-vision-language foundation model OmniVL, to support both image-language and video-language pretraining and corresponding downstream tasks[1], including visual only tasks (*e.g.*, image classification, video action recognition),

---

⋆Work done at Microsoft, currently at Google Brain. †Corresponding authors.

[1]Here we regard models like [71, 70] as image-language only, as they only pretrain on image-language and naively regard video as independent frames without temporal modeling or need heavy adaption to video.

Table 1: A system-level comparison between OmniVL and existing Vision-Languange pretraining and foundation models. "IL","VL" denotes image-language pretraining and video-language pretraining, "Non-Gen" denotes non-generative tasks (*e.g.*, visual only classification, cross-modal alignment), while "Gen" denotes multi-modal generation tasks (*e.g.*, image/video question answering,captioning). "I-L,V-L" and "I-T,V-T" denote image/video-label and image/video-text data respectively.

| Method | Modality Unification | | Functionality Unification | | Data Unification | | | |
|--------|-----|-----|---------|-----|-----|-----|-----|-----|
| | ILP | VLP | Non-Gen | Gen | I-T | I-L | V-T | V-L |
| CLIP [53] | ✓ | | ✓ | | ✓ | | | |
| ALIGN [30] | ✓ | | ✓ | | ✓ | | | |
| VLMO [62] | ✓ | | ✓ | | ✓ | | | |
| ALBEF [38] | ✓ | | ✓ | | ✓ | | | |
| SIMVLM [63] | ✓ | | ✓ | ✓ | ✓ | | | |
| UniVLP [75] | ✓ | | ✓ | ✓ | ✓ | | | |
| BLIP [37] | ✓ | | ✓ | ✓ | ✓ | | | |
| FiT [6] | | ✓ | ✓ | | ✓ | | ✓ | |
| ALPRO [36] | | ✓ | ✓ | | ✓ | | ✓ | |
| VIOLET [23] | | ✓ | ✓ | | ✓ | | ✓ | |
| FLAVA [55] | ✓ | | ✓ | | ✓ | | | |
| Florence [71] | ✓ | | ✓ | | ✓ | ✓ | | |
| OmniVL (Ours) | ✓ | ✓ | ✓ | ✓ | ✓ | ✓ | ✓ | ✓ |

cross-modal alignment tasks (*e.g.*, image/video-text retrieval), and multi-modal understanding and generation tasks (*e.g.*, image/video question answering, captioning) simultaneously. To the best of our knowledge, it is the first time to demonstrate that one model can benefit both image and video tasks *bidirectionally*, as opposed to conventional single directional way, *i.e.*, using image (/image-language) to help video(/video-language).

To support both image and video inputs, OmniVL adopts a unified transformer-based visual encoder to extract visual representations, where video inputs share most transformer layers with images except for the 3D patch tokenizer and temporal attention blocks [8]. Similar to existing vision-language models, OmniVL has another text encoder to extract language representations. To support multiple tasks learning within the same architecture, OmniVL follows an encoder-decoder structure with two visual-grounded decoders. One decoder is designed with bidirectional attention for visual-text semantic alignment, while the other is equipped with causal attention for text generation. We pretrain OmniVL with image-language and video-language data in a *decoupled* joint way, which is different from existing works [37, 70, 71, 53, 72] that apply image-language only pretraining, video-language only pretraining or their joint pretraining from scratch. More specifically, we first pretrain on image-language to focus on spatial representation learning, and then do joint pretraining with video-language together to learn the temporal dynamics incrementally while preserving/polishing the well-learned spatial representations. We believe this not only makes the learning more efficient from spatial to temporal dimensions, but also enforces the learning complementary to each other. This bidirectional help has not been unraveled in prior works, and is important in pushing one foundation model to boost the performance on both image and video tasks.

Moreover, OmniVL is motivated by the unified contrastive learning [69] used in Florence [71], and extends its scope to cover video-text and video-label (*e.g.*, video action recognition) data. The underlying consideration lies in two aspects: 1) As mentioned above, we aim to leverage as much supervised (or noisily supervised) pretraining corpus as possible; 2) As shown in [69], human-annotated visual-label data (*e.g.*, ImageNet [16]) can help to derive more discriminative representations and benefit transfer learning tasks (*e.g.*, image classification), while webly-crawled vision-language data cover broader visual concepts and benefit cross-modal and multi-modal tasks. This simple extension facilitates us to enjoy both advantages.

We call our foundation model OmniVL, since it unifies in three dimensions: modality (*i.e.*, image-language and video-language pretrainings), functionality (*i.e.*, non-generative and generative tasks),

and data unification (*i.e.*, image-text, video-text, image-label and video-label data) as demonstrated in Table 1. With the similar model size and data scale, OmniVL achieves new state-of-the-art or at least competitive results on a wide scope of downstream tasks. For example, when using ViT-Base scale model to pretrain on a moderate data scale (*e.g.*, $\sim$ 14M image-text, $\sim$2.5M video-text), we achieve state-of-the-art performance on image-text retrieval (82.1/64.8 R@1 on COCO for image-to-text/text-to-image), image captioning (39.8 BLEU@4 on COCO), text-to-video retrieval (47.8 R@1 on MSRVTT), and video question answering (51.9% accuracy on MSVD).

## 2 Related Work

**Vision-Only Pretraining.** Large-scale pretraining plays a key role in the success of deep neural networks recently. In the computer vision field, supervised pretraining [27, 26, 33, 18] on ImageNet [16] is the most classical setting. Recently, BiT [33] shows that supervised pretraining on larger-scale datasets with larger models offers better transfer ability. In parallel, self-supervised pretraining has also been extensively studied in the literature, and dominant methods include contrastive learning [12, 40, 25] approaches or BERT-pretraining strategies [19, 7, 61, 20]. Despite their great success, they focus on unimodal pretraining and fail to support cross-modal or multi-modal tasks.

**Vision-Language Pretraining.** Vision-Language pretraining (VLP) [45, 58, 57, 13, 56, 53, 30, 31] has attracted surging attention in the vision-language community, which aims to learn generic multi-modal representations to solve various tasks, *e.g.*, image captioning, image-text retrieval, and video question answering. Depending on the modality of the input data and targeted downstream tasks, existing VLP approaches can be roughly divided into two categories: image-language pretraining methods [13, 42, 75, 63] which learn a joint distribution over visual and linguistic representations from image-text pairs, and video-language methods [39, 35, 36, 23, 6, 49, 4, 2, 51] which model the semantic associations between video frames and texts from video-text pairs. Among them, some recent works [6, 23] also explore image-language and video-language joint pretraining to improve video-language tasks. Instead, OmniVL aims to integrate image-language and video-language within one foundation model. Moreover, inspired by the observation that decoupling spatial and temporal learning is better than direct joint spatial-temporal learning [61, 73], we follow BEVT [61] and introduce a decoupled joint pretraining paradigm, which first learns spatial visual representations with image-language and then conducts joint pretraining. With such a design, we demonstrate for the first time that they can help each other in a bidirectional way. Moreover, as a foundation model, we enable more unification in terms of functionality and pretraining corpus.

**Vision Foundation Models.** Automating the understanding of our multi-modal world with machines requires the development of foundation models that work across different modalities and domains [10, 46]. CLIP [53] and ALIGN [30] are typically regarded as the pioneering explorations of foundation models. By pretraining on web-scale noisy image-text pair data, they excel at cross-modal alignment and zero-shot classification tasks. Florence [71] further extends the scope of foundation models to cover Space-Time-Modality space and performs better especially on vision-only tasks with unified contrastive learning. Despite their success, all the above approaches do not naturally support multi-modal generation tasks (*e.g.*, visual question answering and captioning). To address this limitation, some recent works like FLAVA [55], BLIP [37] and CoCa [70] design one image-language foundation model to support both cross-modal alignment tasks and multi-modal generation tasks. While such image-language foundation models can be extended to support video-language tasks in the fine-tuning stage, they either need heavy task-specific adaptors or simply treat video as independent frames. In contrast, OmniVL is designed to support both image-language and video-language starting from the pretraining stage without any extra adaptors.

## 3 Methodology

### 3.1 Overall Framework

The overall framework of OmniVL is illustrated in Figure 1, which follows an encoder-decoder like structure. OmniVL consists of a unified visual encoder to extract the representations for both images and videos, a text encoder to obtain text representations, and two visual-grounded decoders for semantic alignment and open-ended text generation, respectively. Below we briefly introduce each component and leave the detailed structure in the supplementary material.

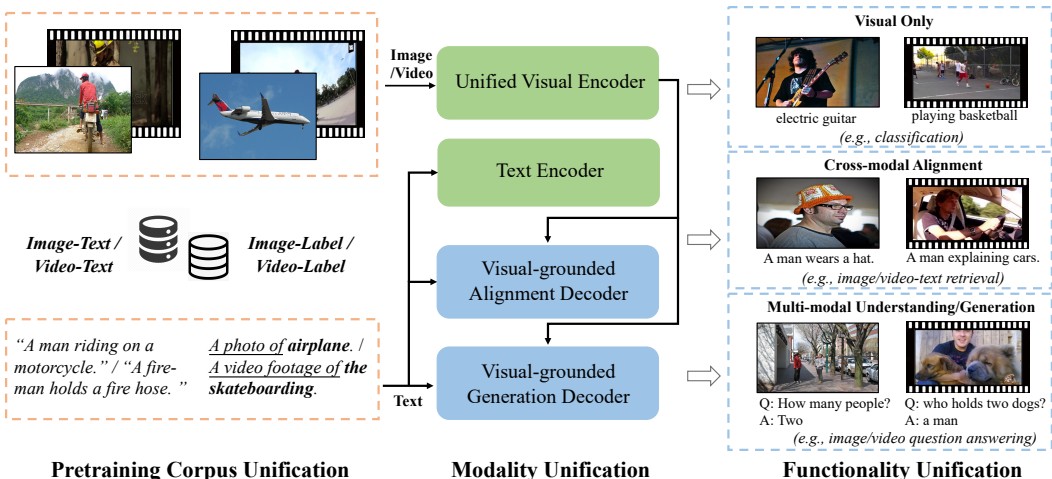

Figure 1: An overview of OmniVL. We unify the pretraining corpus (human-annotated data and webly-crawled data), modality (image, video, and language), and functionality (multi-modal understanding and generation tasks, visual classification tasks) in one universal framework.

**Unified Visual Encoder.** We unify images and videos in a transformer-based visual encoder by converting both of them into a series of tokens, where the independent 2D/3D convolution-based patch tokenizers are used for image/video respectively. Accordingly, spatial and temporal positional encodings are added to the input tokens to incorporate positional information. For the transformer structure, we follow TimeSformer [8] to employ decoupled spatial-temporal attention, which individually models the static spatial appearance and temporal dynamics in visual data. Specifically, within each transformer block, we sequentially perform temporal self-attention and spatial self-attention. The temporal self-attention blocks will be automatically skipped for the image inputs. The final visual representation $v_{cls}$ is obtained from the [CLS] token of the last block. Note that we share the model weights for image and video inputs except for the temporal self-attention.

**Text Encoder.** We adopt BERT [17] as the Text Encoder, which transforms input text into a sequence of token embeddings. The embedding of [CLS] token $w_{cls}$ is used as the language representation.

**Visual-grounded Alignment Decoder.** Even though the above unimodal encoders can support cross-modal alignment like CLIP [53], we employ an extra visual-grounded alignment decoder to further facilitate the learning and enhance the alignment accuracy like [37, 23]. It takes the text and output visual features from the unified visual encoder as input, and fuses the information of both modalities with stacked transformer blocks. Each block basically contains a self-attention layer, a cross-attention layer and a feed-forward layer. Additionally, a task-specific [ENC] token is added to the input text, the output embedding of which will be used as the fused cross-modal representation.

**Visual-grounded Generation Decoder.** We empower our model to own the multi-modal generation capability by attaching a visual-grounded text generation decoder. It adopts the similar architecture to the above alignment decoder, but replaces the bidirectional self-attention with causal self-attention. A [DEC] token and an [EOS] token are added to indicate the task type and signal the end, separately.

## 3.2 Pre-training Objectives

We jointly optimize OmniVL with the following three objectives:

**Unified Vision-Language Contrastive (UniVLC) Loss.** UniCL [69] introduces a novel paradigm for visual representation learning by unifying the supervised learning from image-label data and contrastive learning from the natural language supervision. In this paper, we extend its scope to the unified visual domain, which incorporates both image and video data for cross-modal pretraining via a joint visual-label-text space.

More specifically, we define manually-annotated image/video-label data and web-crawled image/video-text data in a triplet format $\mathcal{S} = (x, y, t)$, where $x \in \mathcal{X}$ is the image/video data,

$y \in \mathcal{Y}$ is the unique label indicating the index of the grouped language description in the whole pretrain dataset, and $t \in \mathcal{T}$ is its corresponding language description. For image/video-label data, $t$ is generated with the same prompt strategy used in CLIP [53] and ActionCLIP [59](*i.e.*, filling the class names into the prompt templates). Note that in this joint visual-label-text space, visual data from manually-annotated dataset belonging to the same category shares the common textual description.

Based on this, given the visual embedding of image/video $x_i$ and the language embedding of its text $t_i$ in a batch $\mathcal{B}$, we follow CLIP to apply a linear projection and normalization layer on them to obtain the latent visual vector $v_i$ and text vector $w_i$. To enjoy a large batch size for contrastive learning, we maintain three memory banks as [25, 38] to store the most recent $M$ visual vectors $\{v_m\}_{m=1}^{M}$ and text vectors $\{w_m\}_{m=1}^{M}$ from the momentum encoders, and the corresponding labels $\{y_m\}_{m=1}^{M}$. Then we calculate the vision-to-text and text-to-vision contrastive loss as:

$$\mathcal{L}_{v2t}(v_i) = -\sum_{k \in \mathcal{P}(i)} \log \frac{\exp(v_i^{\mathrm{T}} w_k)/\tau)}{\sum_{m=1}^{M} \exp(v_i^{\mathrm{T}} w_m/\tau)}, \quad \mathcal{L}_{t2v}(w_i) = -\sum_{k \in \mathcal{P}(i)} \frac{\exp(w_i^{\mathrm{T}} v_k)/\tau)}{\sum_{m=1}^{M} \exp(w_i^{\mathrm{T}} v_m)/\tau)}, \tag{1}$$

where $k \in \mathcal{P}(i) = \{k|k \in M, y_k = y_i\}$, and $\tau$ is a learnable temperature parameter. Finally, the unified vision-language contrastive loss is defined as:

$$\mathcal{L}_{\mathrm{UniVLC}}(; \theta_{ve}, \theta_{te}) = \frac{1}{2}\mathbb{E}_{(x_i, y_i, t_i) \sim (\mathcal{X}, \mathcal{Y}, \mathcal{T})} \left[\mathcal{L}_{v2t}(x_i) + \mathcal{L}_{t2v}(t_i)\right], \tag{2}$$

where $\theta_{ve}$ and $\theta_{te}$ denote the parameters of the unified visual encoder and text encoder.

**Vision-Language Matching (VLM) Loss.** VLM loss encourages the model to learn aligned visual and text representations. Specifically, we randomly replace the text $t_i$ for $x_i$ with the text $t_j$ from a different image/video in the same batch $\mathcal{B}$, and input them to the unified visual encoder and visual-grounded alignment decoder, respectively. Then a linear layer is applied to the output of visual-grounded alignment decoder to produce a two-category probability $p_{vlm}$, which measures whether the input pair is matched. Finally, we optimize the parameters of the unified visual encoder $\theta_{ve}$ and the parameters of visual-grounded alignment decoder $\theta_{ad}$ with VLM loss:

$$\mathcal{L}_{\mathrm{VLM}}(; \theta_{ve}, \theta_{ad}) = \mathbb{E}_{(x_i, y_i, t_i) \sim (\mathcal{X}, \mathcal{Y}, \mathcal{T})} \left[y_{vlm}\log p_{vlm} + (1 - y_{vlm})\log(1 - p_{vlm})\right], \tag{3}$$

where $y_{vlm} = 1$ if $j \in \mathcal{B}$ and $y_j = y_i$, otherwise $y_{vlm} = 0$.

**Language Modeling (LM) Loss.** Previous works indicate that LM facilitates the model to develop better text-induced generalization ability [63]. Therefore, we optimize the output of visual-grounded generation decoder with a cross-entropy loss, which directly maximizes the likelihood of the input text sequence in an autoregressive manner:

$$\mathcal{L}_{\mathrm{LM}}(; \theta_{ve}, \theta_{gd}) = -\mathbb{E}_{(x_i, y_i, t_i) \sim (\mathcal{X}, \mathcal{Y}, \mathcal{T})} \left[\sum_{l=1}^{L} \log P(t_i^l | t^{<l}, x_i)\right]. \tag{4}$$

where $L$ is the length of the input text, $\theta_{ve}$ and $\theta_{gd}$ represent the parameters of the unified visual encoder and visual-grounded generation decoder.

Combining Eqn. 2-Eqn. 4, the overall objectives can be summarized as:

$$\mathcal{L} = \lambda_1 \mathcal{L}_{\mathrm{UniVLC}} + \lambda_2 \mathcal{L}_{\mathrm{VLM}} + \lambda_3 \mathcal{L}_{\mathrm{LM}}. \tag{5}$$

where $\lambda_1$, $\lambda_2$, and $\lambda_3$ are weighting hyper-parameters and all set as 1 by default.

### 3.3 Pretraining Corpus and Paradigms

**Corpus.** As mentioned before, our pretraining corpus includes both visual-text data and visual-label data, benefiting from the unified visual-label-text space. For the image-text data, we adopt the same pre-training dataset as [38, 37] with 14M images in total by default, including two human-annotated datasets (COCO [43] and Visual Genome [34]), and three web datasets (CC3M [54], CC12M [11], and SBU captions [50]). For the video-text data, we use WebVid [6] which contains 2.5M videos from the web. The visual-label datasets that we adopt includes the image dataset ImageNet-1K [16] and video dataset Kinetics-400 [32]. As some baseline image-language methods only use 4M image-text data by excluding CC12M, we also provide the corresponding results in the experiment part.

Table 2: Linear probing evaluation on 6 image classification datasets.

| Method | Img-text pairs | Food101 | CIFAR10 | CIFAR100 | Pets | DTD | Flowers | Avg |
|---|---|---|---|---|---|---|---|---|
| METER-CLIP-B/16 [22] | 400M+4M | 79.2 | 91.8 | 70.3 | 40.4 | 62.2 | 67.1 | 68.5 |
| ALBEF [38] | 14M | 84.0 | 95.6 | 80.8 | 68.4 | 73.4 | 86.5 | 81.4 |
| BLIP [37] | 14M | 85.5 | 95.2 | 80.0 | 65.3 | 74.6 | 88.4 | 81.5 |
| FLAVA [55] | 14M | 85.2 | 90.4 | 76.2 | 82.3 | 74.2 | 92.7 | 83.5 |
| FLAVA [55] | 70M | 88.5 | 92.9 | 77.7 | 84.8 | 77.3 | **96.4** | 86.3 |
| CLIP-ViT-B/16 | 400M | **92.8** | 96.2 | 83.1 | 86.7 | **79.2** | 93.1 | **88.5** |
| OmniVL | 14M$^*$ | 87.4 | **96.2** | **83.2** | **87.1** | 76.5 | 89.8 | 86.7 |

**Paradigms.** In contrast to some existing methods [6, 23] that conduct joint pretraining on image data and video data from scratch, we adopt a *decoupled* joint pretraining paradigm instead. Specifically, we first pretrain our model on image-label-text data, and then perform joint training on both image-label-text data and video-label-text data. In this way, we decouple the multi-modal modeling into spatial and temporal dimensions. Such a design has two potential benefits: 1) Considering the expensive computational cost of video pretraining, applying the image data to learn the spatial representation first is more efficient. 2) The decoupled pattern makes the multimodal representation learning more effective, which is the key to make image-language and video-language benefit each other.

## 4 Experiments

**Implementation Details.** By default, we use the TimeSformer base model and BERT base model for visual encoder and text encoder, respectively. As mentioned in Sec 3.3, our pretraining follows a *decoupled* paradigm. For the image-language pretraining stage, we initialize spatial attention with ViT-B/16 [21] pretrained on ImageNet-1K [16]. We take random image crops of resolution $224 \times 224$ as inputs and apply RandAugment [15]. The model is pretrained for 20 epochs using a batch size of 2880. For the joint pretraining, we sparsely sample $8 \times 224 \times 224$ video clips, and train the model for 10 epochs with a batch size of 800 for video data and 2880 for image data. Our joint pretraining alternates batches between the image and video data. The model is optimized with AdamW [44] using a weight decay of 0.05. The learning rate is warmed-up to 3e-4 (image) / 8e-5 (joint) and decayed linearly with a rate of 0.85. During downstream fine-tuning, we increase the image resolution to $384 \times 384$ for both image-text and video-text tasks [38, 37], unless otherwise specified. We randomly sample 8 frames per video for retrieval and 16 for QA, following [36]. Temporal position embeddings in the spatial-temporal visual encoder are interpolated to accommodate the inputs of different lengths. Besides, we use task-specific learning rates and training epochs due to various data scales and domains, the details of which will be further illustrated in the supplementary material.

### 4.1 Visual Only Tasks

We first evaluate the representations of the visual encoder on visual only tasks. Here the classical image classification and video action recognition tasks are adopted for benchmarking. Note that, in the following tables, *we use the superscript "*" to denote extra video data is used*.

**Image Classification.** Linear probing is a commonly used method to evaluate the representation quality [53, 71, 55]. Following the implementation of CLIP [53], we freeze the visual encoder and fine-tune the newly appended linear layers for linear probing. We evaluate our method on 6 image classification datasets, which are not included in our pretraining set. We compare with METER [22], ALBEF [38], BLIP [37], FLAVA [55], and CLIP [53] in Table 2. Note that for fair comparisons with FLAVA [55], we pretrain their model (we adopt the implementation in torchmultimodal[2]) on the 14M image-text data. Compared to METER, ALBEF, BLIP, and FLAVA$_{14M}$, OmniVL offers consistently better results. Compared to CLIP and FLAVA$_{70M}$, even though we use far less pre-training data, our method still achieves overall comparable results.

**Video Action Recognition.** Video action recognition is one of the most representative tasks for video understanding [80]. We first report the linear probing results on UCF101 and HMDB51. The results are summarized in the first two columns of Table 3. We see that OmniVL achieves

---

[2]https://github.com/facebookresearch/multimodal

Table 3: Comparison with baseline methods on video action recognition datasets Kinetics-400 and Something-Something v2 under fine-tuning settings, and UCF101 and HMDB51 under linear probing settings. "Sup21K" denotes supervised pretraining on ImageNet-21 dataset.

| Method | UCF101 | HMDB51 | K400 | | SSV2 | |
| | Top-1 | Top-1 | Top-1 | Top-5 | Top-1 | Top-5 |
|---|---|---|---|---|---|---|
| TimeSformer [8]-Sup21K | 82.9 | 60.1 | 78.0 | 93.7 | 59.5 | - |
| FiT [6] | 89.6 | 68.8 | 78.5 | 94.1 | 61.6 | 85.7 |
| OmniVL | **93.2** | **70.0** | **79.1** | **94.5** | **62.5** | **86.2** |

Table 4: Fine-tuned image-text retrieval results on Flickr30K and COCO datasets. We report text recall@1 / recall@5 / recall@10, and image recall@1 / recall@5 / recall@10.

| Method | # Img-Text Pairs | COCO (5K test set) | | | | | | Flickr30K (1K test set) | | | | | |
| | | TR | | | IR | | | TR | | | IR | | |
|---|---|---|---|---|---|---|---|---|---|---|---|---|---|
| VirTex [48] | - | - | - | - | 38.1 | 62.8 | - | - | - | - | 35.1 | 64.6 | - |
| UNITER [13] | 4M | 65.7 | 88.6 | 93.8 | 52.9 | 79.9 | 88.0 | 87.3 | 98.0 | 99.2 | 75.6 | 94.1 | 96.8 |
| OSCAR [42] | 4M | 70.0 | 91.1 | 95.5 | 54.0 | 80.8 | 88.5 | - | - | - | - | - | - |
| UNIMO [41] | 4M | - | - | - | - | - | - | 89.4 | 98.9 | 99.8 | 78.0 | 94.2 | 97.1 |
| VLMO [62] | 4M | 74.8 | 93.1 | 96.9 | 57.2 | 82.6 | **89.8** | 92.3 | 99.4 | 99.9 | 79.3 | 95.7 | 97.8 |
| OmniVL | 4M* | **76.8** | **93.6** | **97.3** | **58.5** | 82.6 | 89.5 | **94.9** | **99.6** | **99.9** | **83.4** | **97.0** | **98.6** |
| FLAVA [55] | 70M | 61.5 | 82.1 | 89.6 | 50.1 | 74.4 | 83.2 | 85.4 | 95.7 | 98.3 | 73.2 | 92.7 | 95.5 |
| METER [22] | 404M | 76.2 | 93.2 | 96.8 | 57.1 | 82.7 | 90.1 | 94.3 | 99.6 | 99.9 | 82.2 | 96.3 | 98.4 |
| ALIGN [30] | 1.8B | 77.0 | 93.5 | 96.9 | 59.9 | 83.3 | 89.8 | 95.3 | 99.8 | 100.0 | 84.9 | 97.4 | 98.6 |
| ALBEF [38] | 14M | 77.6 | 94.3 | 97.2 | 60.7 | 84.3 | 90.5 | 95.9 | 99.8 | 100.0 | 85.6 | 97.5 | 98.9 |
| BLIP [37] | 14M | 80.6 | 95.2 | 97.6 | 63.1 | 85.3 | 91.1 | 96.6 | 99.8 | 100.0 | 87.2 | 97.5 | 98.8 |
| Florence [71] | 900M | 81.8 | 95.2 | - | 63.2 | 85.7 | - | 97.2 | 99.9 | - | 87.9 | **98.1** | - |
| OmniVL | 14M* | **82.1** | **95.9** | **98.1** | **64.8** | **86.1** | **91.6** | **97.3** | **99.9** | **100.0** | 87.9 | 97.8 | **99.1** |

excellent performance, *i.e.*, 93.2% on UCF101, even without end-to-end training, which beats baseline methods by a large margin. Furthermore, we conduct fine-tuning experiments on Kinetics-400 [32] and Something-Something v2 [24] dataset. We compare our method with supervised pretrained TimeSformer [8] and FiT [6] in Table 3 since we share the same model architecture. The same training settings with TimeSformer are adopted for fair comparisons. OmniVL outperforms TimeSformer by 1.4% and 5.0% on Kinetics-400 and Something-Something v2 in terms of Top-1 accuracy, respectively. Compared to the video-language model FiT [6] (*i.e.*, performing joint pretraining from scratch), our results are also overall better, highlighting the advantage of our method.

## 4.2 Cross-modal Alignment Tasks

Using visual and text embeddings generated from the unimodal encoders, OmniVL can easily handle cross-modal alignment tasks, *e.g.*, image-text retrieval and text-to-video retrieval. In addition, in order to balance the inference efficiency and the deep fusion of multi-modal information, we follow [37, 38] to first select Top-K ($K = 128$ by default) candidates based on the vision-language similarity scores, which are further re-ranked by calculating their pairwise VLM scores. The pretrained model is fine-tuned with UniVLC loss and VLM loss.

**Image-Text Retrieval.** We first evaluate OmniVL on COCO [43] and Flickr30K [52] for both image-to-text retrieval and text-to-image retrieval. As shown in Table 4, OmniVL outperforms other methods by clear margins. With 14M image-text pairs for pretraining, our model surpasses BLIP by 1.8% and 0.8% in terms of average recall@1 on COCO and Flickr30K, respectively, which is even competitive compared with Florence that is trained with much larger scale data and model .

**Text-to-Video Retrieval.** We compare OmniVL with other methods on MSRVTT [67] and DiDeMo [5] for text-to-video retrieval under both fine-tuning and zero-shot transfer settings in Table 5. Note that directly using the pretrained models, OmniVL achieves 42.0% and 40.6% on MSRVTT and DiDeMo in terms of recall@1, respectively, significantly surpassing existing methods.

Table 5: Comparison with SOTA text-to-video-retrieval methods on MSRVTT and DiDeMo with fine-tune (left) and zero-shot (right) evaluation. R@1 / R@5 / R@10 are reported.

| Method | Text-to-Video Retrieval | | | | | | Zero-shot Retrieval | | | | | |
| | MSRVTT | | | DiDeMo | | | MSRVTT | | | DiDeMo | | |
|---|---|---|---|---|---|---|---|---|---|---|---|---|
| ClipBERT [35] | 22.0 | 46.8 | 59.9 | 20.4 | 48.0 | 60.8 | - | - | - | - | - | - |
| TT-CE+ [14] | 29.6 | 61.6 | 74.2 | 21.6 | 48.6 | 62.9 | - | - | - | - | - | - |
| VideoCLIP [66] | 30.9 | 55.4 | 66.8 | - | - | - | 10.4 | 22.2 | 30.0 | 16.6 | 46.9 | - |
| FiT [6] | 32.5 | 61.5 | 71.2 | 31.0 | 59.8 | 72.4 | 18.7 | 39.5 | 51.6 | 21.1 | 46.0 | 56.2 |
| TT-CE+ (+QB-NORM) [9] | 33.3 | 63.7 | 76.3 | 24.2 | 50.8 | 64.4 | - | - | - | - | - | - |
| ALPRO [36] | 33.9 | 60.7 | 73.2 | 35.9 | 67.5 | 78.8 | 24.1 | 44.7 | 55.4 | 23.8 | 47.3 | 57.9 |
| VIOLET [23] | 34.5 | 63.0 | 73.4 | 32.6 | 62.8 | 74.7 | 25.9 | 49.5 | 59.7 | 23.5 | 49.8 | 59.8 |
| OmniVL | **47.8** | **74.2** | **83.8** | **52.4** | **79.5** | **85.4** | **34.6** | **58.4** | **66.6** | **33.3** | **58.7** | **68.5** |

Table 6: Comparison with state-of-the-art image captioning methods on NoCaps and COCO Caption. C: CIDEr, S: SPICE, B@4: BLEU@4.

| Method | # Img-Text Pairs | NoCaps | | | | | | | | COCO Caption Karpathy test | |
| | | in-domain | | near-domain | | out-domain | | overall | | | |
| | | C | S | C | S | C | S | C | S | B@4 | C |
|---|---|---|---|---|---|---|---|---|---|---|---|
| Enc-Dec [11] | 15M | 92.6 | 12.5 | 88.3 | 12.1 | 94.5 | 11.9 | 90.2 | 12.1 | - | 110.9 |
| VinVL [74] | 5.7M | 103.1 | 14.2 | 96.1 | 13.8 | 88.3 | 12.1 | 95.5 | 13.5 | 38.2 | 129.3 |
| LEMON [29] | 12M | 104.5 | 14.6 | 100.7 | 14.0 | 96.7 | 12.4 | 100.4 | 13.8 | - | - |
| BLIP [37] | 14M | **111.3** | **15.1** | 104.5 | 14.4 | 102.4 | 13.7 | 105.1 | 14.4 | 38.6 | 129.7 |
| SIMVLM [63] | 1.8B | - | - | - | - | - | - | 94.8 | 13.1 | 39.0 | 134.8 |
| OFA$_{14M}$ [60] | 14M | - | - | - | - | - | - | - | - | 38.7 | 130.5 |
| OFA [60] | 21.4M | - | - | - | - | - | - | - | - | **41.0** | **138.2** |
| OmniVL | 14M* | 104.6 | 15.0 | **108.3** | **14.9** | **106.3** | **14.2** | **107.5** | **14.7** | 39.8 | 133.9 |

The performance of OmniVL is further improved under the fine-tuning settings. The results suggest the multi-modal representations learned by our method are very discriminative.

## 4.3 Multi-modal Understanding and Generation Tasks

The visual-grounded generation decoder equips OmniVL with the capability for multi-modal understanding and generation so as to reason and describe. In this part, we fine-tune our model with the LM loss, and evaluate its performance on captioning and image/video question answering tasks.

**Image Captioning.** Image captioning requires the model to generate a textual description for a given image. We fine-tune our model on COCO and then evaluate on both COCO [43] NoCaps [1]. Following [63, 37], we adopt a prefix prompt "a picture of" to guide the caption generation. The results are presented in Table 6, from which we can see that OmniVL achieves superior results on both datasets, *e.g.*, 107.5 and 133.9 on NoCaps and COCO in terms of CIDEr. Although SimVLM [63] adopt much larger pretraining data than ours, OmniVL still achieves comparable or even better results on some metrics, *e.g.*, CIDEr, on COCO dataset. For fair comparison with OFA [60], we pre-train their model on the 14M image-text data (with only image-text matching and image captioning objectives, denoted as OFA$_{14M}$), the results demonstrate that using the same amount of pre-training data, OmniVL performs better than OFA measured by all the metrics. Note that we don't show the results of SIMVLM on 14M data since they haven't released their code.

**Video Captioning.** In this section, we evaluate OmniVL on YouCook2 [76] for video captioning. We follow [77] to report the results on the validation sets in Table 7. We can see that OmniVL outperforms most existing methods in all the metrics. The models marked in gray, *e.g.*, UniVL [47], apply an extra pre-trained backbone network S3D [64] to extract video-level features offline. Comparatively, OmniVL is fine-tuned in an end-to-end manner.

**Visual Question Answering.** For VQA, the model is expected to predict an answer given an image and a related question. To this end, we follow [38, 37] to formulate it as a generation task and focus on open-ended VQA. It is worth noting that during fine-tuning, we first input the image and the question into the unified visual encoder and visual-grounded alignment decoder separately to obtain a fused

Table 7: Comparison with SOTA methods on Youcook2 dataset for video captioning. The results in gray denote using pretrained backbones to extract video-level features.

| Method | B@3 | B@4 | METEOR | ROUGE-L | CIDEr |
|---|---|---|---|---|---|
| Bi-LSTM [76] | - | 0.87 | 8.15 | - | - |
| EMT [77] | - | 4.38 | 11.55 | 27.44 | 0.38 |
| VideoBERT [57] | 6.80 | 4.04 | 11.01 | 27.50 | 0.49 |
| ActBERT [78] | 8.66 | 5.41 | 13.30 | 30.56 | 0.65 |
| AT [28] | - | 8.55 | 16.93 | 35.54 | 1.06 |
| UniVL [47] | 16.46 | 11.17 | 17.57 | 40.09 | 1.27 |
| OmniVL | **12.87** | **8.72** | **14.83** | **36.09** | **1.16** |

Table 8: Comparison with SOTA methods on VQA for visual question answering.

| Method | # Img-Text Pairs | test-dev | test-std |
|---|---|---|---|
| FLAVA [55] | 68M | 72.80 | - |
| OSCAR [42] | 4M | 73.16 | 73.44 |
| ALBEF [38] | 14M | 75.84 | 76.04 |
| BLIP [37] | 14M | 77.54 | 77.62 |
| METER [22] | 404M | 77.68 | 77.64 |
| SimVLM [63] | 1.8B | 77.87 | 78.14 |
| OFA [60] | 21.4M | 78.00 | 78.10 |
| OmniVL | 14M* | **78.33** | **78.35** |

Table 9: Accuracy (%) of video question answering on MSRVTT and MSVD.

| Method | MSRVTT | MSVD |
|---|---|---|
| ClipBERT [35] | 37.4 | - |
| JustAsk [68] | 41.5 | 46.3 |
| ALPRO [36] | 42.1 | 45.9 |
| MERLOT [72] | 43.1 | - |
| VIOLET [23] | 43.9 | 47.9 |
| OmniVL | **44.1** | **51.0** |

multi-modal representation, and then feed the visual representation and multi-modal representation into the visual-grounded generation decoder to predict the final answer. We compare with existing SOTA methods in Table 8. We observe consistent performance gain compared with existing methods. Especially, OmniVL even outperforms SimVLM [63] by 0.6% trained with 1.8B image-text pairs.

**Video Question Answering.** Table 9 summarizes the results of video question answering on MSRVTT-QA [65] and MSVD-QA [65]. OmniVL surpasses both QA-specific methods, *e.g.*, JustAsk [68], and pretraining methods, *e.g.*, VIOLET [23] on both datasets, which validates the effectiveness of our method for complex multimodal modeling.

## 4.4 Ablation Study

**Decoupled Joint Pretraining.** To verify the effect of decoupled joint pretraining, we conduct four ablation experiments with different pretraining strategies: image-only pretraining, video-only pretraining, joint pretraining from scratch, and Img2Vid pretraining where we first pretrain OmniVL on image and then on video. We list some representative results of each task in Table 10 (see data details and more results in the supplementary material).

Table 10: Comparison results on various downstream tasks by using image/video-only pretraining, joint pretraining from scratch, and decoupled joint pretraining.

| Pretraining | COCO (ret) | | MSRVTT (ret) | COCO (caption) | | VQA | MSRVTT(QA) |
|---|---|---|---|---|---|---|---|
| | TR@1 | IR@1 | IR@1 | B@4 | C | test-dev | acc |
| Without Pretraining | 37.1 | 28.5 | 9.6 | 27.4 | 80.0 | 39.51 | 36.6 |
| Video-only | - | - | 13.7 | - | - | - | 15.8 |
| Image-only | 80.9 | 63.0 | 38.2 | 39.3 | 131.6 | 77.62 | 40.8 |
| Joint | 50.2 | 35.0 | 23.6 | 29.7 | 94.6 | 47.78 | 38.8 |
| Img2Vid | 79.7 | 61.8 | 42.5 | 38.6 | 129.5 | 77.43 | 42.8 |
| Decoupled Joint (ours) | **82.1** | **64.8** | **47.8** | **39.8** | **133.9** | **78.33** | **44.1** |

We can see that video-only pretraining leads to significant performance degradation due to limited data scale for pretraining. Comparatively, image-only pretraining is a competitive baseline, which however, is still far behind decoupled joint pretraining on video tasks, *e.g.*, text-video-retrieval

MSRVTT (ret) and video question answering MSRVTT (QA). Compared to video-only pretraining, joint pretraining from scratch can significantly improve the performance on video tasks, which has also been verified in [6, 23]. However, it produces limited results on image tasks, which is even worse than image-only pretraining. Img2Vid is another competitive baseline, which however demonstrates degraded performance on image tasks compared to image-only pretraining. This indicates the naive combination of image-language and video-language cannot enjoy their synergy.

By contrast, our simple decoupled joint pretraining strategy can achieve better performance on both image and video tasks. We hypothesize this is because starting from image-only pretraining can help the model focus on spatial representation learning first and provides better initialization, and thus the subsequent joint pretraining would be more concentrated on learning the temporal dynamics incrementally while preserving/polishing the well-learned spatial representations.

**UniVLC Loss.** We further replace the UniVLC loss with vanilla contrastive loss to study its impact on various downstream tasks. Note that we exclude the visual-label data from the pretraining corpus under the "w/o UniVLC" setting. The example results shown in Figure 2 illustrate that our method performs comparably to vanilla contrast-based model on vision-language tasks, *e.g.*, image/video-text retrieval, image captioning, and image/video question answering. But on visual only tasks, *e.g.*, linear probing for image/video classification and video action recognition fine-tuning, the performance gain is much higher, indicating that UniVLC could facilitate model to learn more discriminative visual representations and benefit transfer learning tasks.

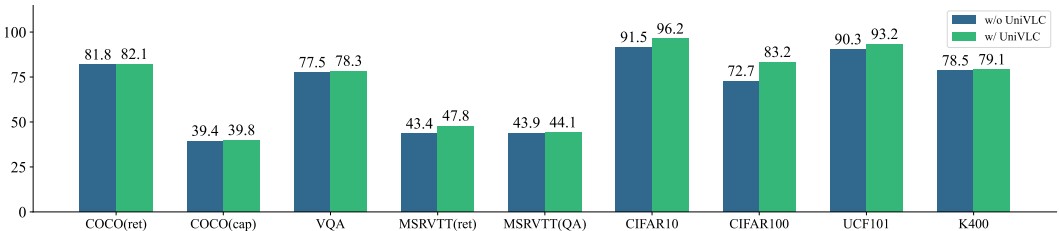

Figure 2: Evaluation on different tasks w/ and w/o UniVLC. We report the text recall@1 on COCO retrieval, B@4 on COCO captioning, test-dev on VQA, recall@1 on MSRVTT(ret), accuracy on MSRVTT(QA), accuracy on CIFAR10, CIFAR100, UCF101 (linear probe), and K400(fine-tuned).

## 5 Conclusion and Discussion of Broader Impact

In this paper, we presented OmniVL, a novel vision-language foundation model that unifies image-language and video-language. It naturally supports visual only tasks, cross-modal alignment tasks, and multi-modal understanding and generation tasks. The unified vision-language contrastive loss also enables OmniVL to utilize image-text, image-label, video-text and video-label data together. Accordingly, a *decoupled* pretraining paradigm is introduced to decouple vision-language modeling into spatial and temporal dimensions, which boosts the performance on both image and video tasks.

Although our model has achieved superior results on a wide range of downstream tasks, it still lacks of the commonsense reasoning capability required by some visual-language interaction tasks (e.g., visual/video question answering). It also needs better architecture design to support the zero-shot capability for visual question answering and few-shot task customization capability like GPT-3. From the societal impact perspective, since our model is pretrained on the large-scale web-crawled data which may contain some toxic language or bias, and it is not easy to explicitly control the model output, much attention should be paid to ensure responsible model deployment.

**Acknowledgement** This project was partially supported by NSFC under Grant No. 62102092. Y.-G. Jiang was sponsored in part by "Shuguang Program" supported by Shanghai Education Development Foundation and Shanghai Municipal Education Commission (No. 20SG01).

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
