# OmniVL: One Foundation Model for Image-Language and Video-Language Tasks
## *Supplementary Material*

## A    Specification for the Visual-grounded Alignment / Generation Decoder

As mentioned in the paper, the visual-grounded alignment decoder is applied to enable the deep interaction of multimodal information with cross-attention blocks, while the visual-grounded generation decoder is adopted to generate natural languages conditioned on the visual input. We further specify their architectures in Figure 1.

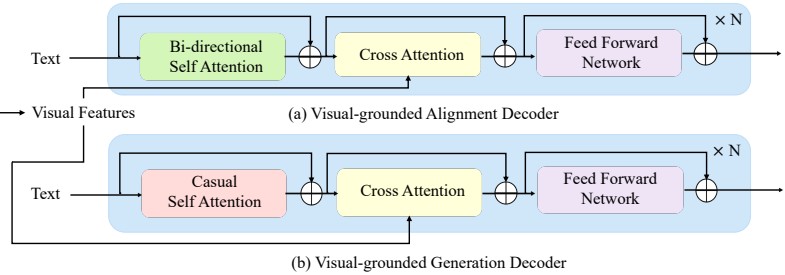

Figure 1: Architecture of the visual-grounded alignment / generation decoder.

Note that both visual-grounded alignment decoder and visual-grounded generation decoder are initialized with the Bert-base model [2], which stacks 12 transformer layers.

## B    Image / Video Question Answering

Image / video question answering requires the model to answer a question according to a given image / video, which models the complex interaction between visual and linguistic representations. During finetuning, we rearrange the pre-trained model, as shown in Figure 2.

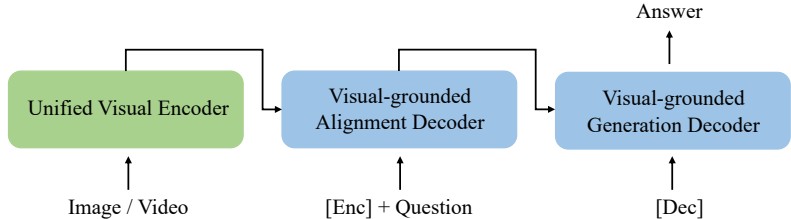

Figure 2: Architecture of the visual-grounded alignment / generation decoder.

Our setup is based on the following considerations. We first input the image / video to unified visual encoder, the output of which will be combined with the text features of the questions through the

36th Conference on Neural Information Processing Systems (NeurIPS 2022).

visual-grounded alignment decoder. Based on these deeply fused representations, we finally generate the predicted answers with the visual-grounded generation decoder.

## C   Finetuning Setups

In this section, we describe the settings used when fine-tuning the pretrained models on various downstream tasks.

### C.1   Image-Language Tasks

For image-text retrieval and image captioning, we resize the images to $384 \times 384$, while for visual question answering, we resize the images to $480 \times 480$, following [4]. We use RandomAugment [1] for data augmentation. The default settings for finetuning on each dataset are shown in Table 1.

Table 1: End-to-end finetuning configurations for image-language downstream tasks.

| Config | COCO (retrieval) & Flickr30k | COCO (captioning) | VQA |
|---|---|---|---|
| optimizer | AdamW | AdamW | AdamW |
| base learning rate | 1e-5 | 1e-5 | 2e-5 |
| weight decay | 0.05 | 0.05 | 0.05 |
| learning rate schedule | linear decay | linear decay | linear decay |
| batch size | 512 | 512 | 256 |
| training epochs | 10 | 10 | 10 |

### C.2   Video-Language Tasks

For all video-language downstream tasks, we resize video frames to $384 \times 384$. During fine-tuning, we randomly sample N frames from each video, where N = 8 for text-to-video retrieval, N = 16 for video question answering following [3], and N = 24 for video captioning. We perform uniform sampling during inference. Similar with image-language tasks, we also adopt RandomAugment [1] for data augmentation. The default settings for finetuning on each dataset are shown in Table 2.

Table 2: End-to-end finetuning configurations for video-language downstream tasks.

| Config | MSRVTT (ret) | DiDeMo | MSRVTT (QA) | MSVD (QA) | Youcook2 |
|---|---|---|---|---|---|
| optimizer | AdamW | AdamW | AdamW | AdamW | AdamW |
| base lr | 5e-6 | 1e-5 | 5e-6 | 1e-5 | 1e-5 |
| weight decay | 0.05 | 0.05 | 0.05 | 0.05 | 0.05 |
| lr schedule | linear decay | linear decay | linear decay | linear decay | linear decay |
| batch size | 32 | 32 | 32 | 32 | 32 |
| training epochs | 6 | 6 | 10 | 10 | 10 |

## D   More Comparison Results on Vision-language Tasks for Different Pretraining Paradigms

We demonstrate more comparison results using different pretraining paradigms (*i.e.*, image-only, video-only, joint pretraining from scratch, and our decoupled pretraining) on various vision-language downstream tasks in Table 3. Details of the pretraining data can be found in Table 4. Moreover, an "img2vid" strategy is also adopted for further comparison, where we start with image-only pretraining and then implement video-only pretraining. We can see our decoupled joint pretraining paradigm achieves consistently better results on all the downstream tasks.

Table 3: More comparison results on various vision-language tasks for different paradigms.

| Method | COCO (5K test set) | | | | | | Flickr30K (1K test set) | | | | | |
|---|---|---|---|---|---|---|---|---|---|---|---|---|
| | TR | | | IR | | | TR | | | IR | | |
| Image-only | 80.9 | 94.8 | 97.5 | 63.2 | 85.2 | 91.3 | 96.6 | 99.8 | 100.0 | 87.2 | 97.5 | 98.8 |
| Joint | 50.2 | 75.6 | 84.9 | 35.0 | 62.7 | 73.9 | 67.2 | 83.4 | 92.1 | 56.5 | 63.4 | 71.7 |
| Img2Vid | 79.7 | 94.8 | 97.7 | 61.8 | 84.7 | 90.9 | 95.8 | 99.6 | 99.9 | 76.5 | 97.3 | 98.2 |
| Decoupled Joint | **82.1** | **95.9** | **98.1** | **64.8** | **86.1** | **91.6** | **97.3** | **99.9** | **100.0** | **87.9** | 97.8 | **99.1** |

| Method | Text-to-Video Retrieval | | | | | | Zero-shot Retrieval | | | | | |
|---|---|---|---|---|---|---|---|---|---|---|---|---|
| | MSRVTT | | | DiDeMo | | | MSRVTT | | | DiDeMo | | |
| Video-only | 13.7 | 33.5 | 41.9 | 18.2 | 43.6 | 52.5 | 6.7 | 19.4 | 29.4 | 7.1 | 18.1 | 27.8 |
| Joint | 23.6 | 49.7 | 61.5 | 28.1 | 52.8 | 64.4 | 15.5 | 39.6 | 53.4 | 19.2 | 42.7 | 51.9 |
| Img2Vid | 42.5 | 71.3 | 79.9 | 51.1 | 76.6 | 82.8 | 38.3 | 56.1 | 64.4 | 37.5 | 62.0 | 72.6 |
| Decoupled Joint | **47.8** | **74.2** | **83.8** | **52.4** | **79.5** | **85.4** | **42.0** | **63.0** | **73.0** | **40.6** | **64.6** | **74.3** |

| Method | NoCaps | | | | | | | | COCO Caption Karpathy test | |
|---|---|---|---|---|---|---|---|---|---|---|
| | in-domain | | near-domain | | out-domain | | overall | | | |
| | C | S | C | S | C | S | C | S | B@4 | C |
| Image-only | 100.2 | 14.4 | 107.2 | 14.6 | 102.7 | 13.8 | 105.5 | 14.4 | 39.3 | 131.6 |
| Joint | 100.0 | 14.1 | 95.7 | 13.6 | 77.4 | 11.6 | 93.0 | 13.4 | 29.6 | 94.6 |
| Img2Vid | 99.2 | 14.1 | 102.7 | 14.2 | 98.5 | 13.4 | 101.5 | 14.0 | 38.6 | 129.5 |
| Decoupled Joint | 104.6 | 15.0 | **108.3** | **14.9** | **106.3** | **14.2** | **107.5** | **14.7** | **39.8** | **133.9** |

| Method | test-dev | test-std |
|---|---|---|
| Image-only | 77.55 | 77.53 |
| Joint | 47.78 | 47.80 |
| Img2Vid | 77.43 | 77.48 |
| Decoupled Joint | **78.33** | **78.35** |

| Method | MSRVTT | MSVD |
|---|---|---|
| Video-only | 15.8 | 17.3 |
| Joint | 38.8 | 39.2 |
| Img2Vid | 42.8 | 48.3 |
| Decoupled Joint | **44.1** | **51.0** |

| Method | B@4 | C |
|---|---|---|
| Video-only | 3.56 | 0.29 |
| Joint | 4.47 | 0.55 |
| Img2Vid | 7.80 | 1.05 |
| Decoupled Joint | **8.72** | **1.16** |

Table 4: Pretraining data used for different pretraining paradigms.

| Method | Image-Text | Image-Label | Video-Text | Video-Label | |
|---|---|---|---|---|---|
| Video-only | - | - | 2.5M | 0.3M | |
| Image-only | 14M | 1.3M | - | - | |
| Joint | 14M | 1.3M | 2.5M | 0.3M | |
| Img2Vid | 14M | 1.3M | 2.5M | 0.3M | |
| Decoupled Joint | 14M | 1.3M | 2.5M | 0.3M | |

# E    Image/Video Captioning Examples

We show some image and video captioning results generated by our method in Figure 3 and Figure 4, respectively. We can see that the captions generated by OmniVL are both natural and abundant. Specifically, for the image captioning, when the visual information in the images is relatively simple, the generated captions are relatively general (line 2 and line 3). While when the contents are rich, OmniVL can generate more fine-grained descriptions (line 1). Fo video captioning, OmniVL could accurately describe the actions (*e.g.*, "add" and "pour") and objects (*e.g.*, "lemon juice" and "fried chicken") in videos. The visualization results demonstrate the superior multimodal generation capability of OmniVL.

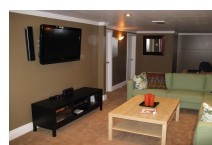 a living room filled with furniture and a flat screen tv.

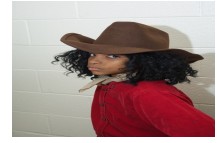 a woman wearing a brown hat and a red shirt.

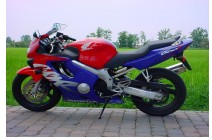 a red and blue motorcycle parked in front of a grassy field.

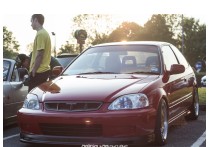 a man standing next to a red car in a parking lot.

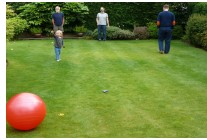 a group of people standing on top of a lush green.

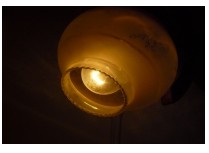 a light that is shining in the dark.

Figure 3: Some captions generated by OmniVL.

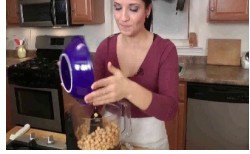 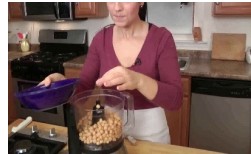 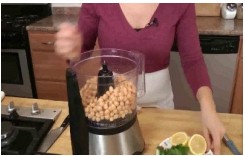 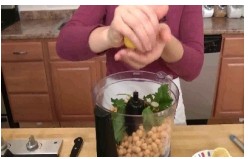

add chickpeas parsley and lemon juice to the food processor and blend

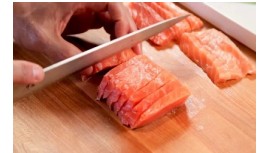 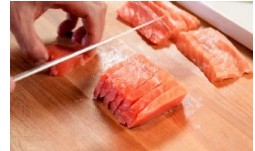 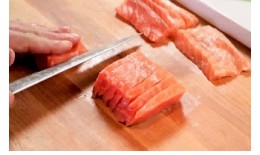 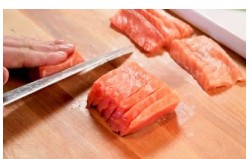

cut the salmon into thin slices

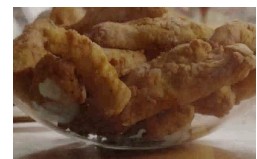 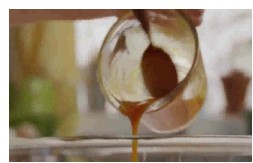 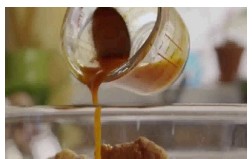 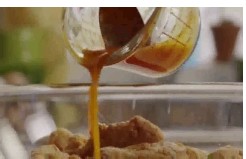

pour the sauce on the fried chicken

Figure 4: Some video captions generated by OmniVL.