# OpenReview forum: "OmniVL: One Foundation Model for Image-Language and Video-Language Tasks"
_NeurIPS.cc/2022/Conference — NeurIPS 2022 Accept_

### Official Review · Reviewer_VLaG · 2022-07-10

**Rating:** 6
**Confidence:** 4
**Soundness:** 3 good
**Presentation:** 3 good
**Contribution:** 3 good

**Summary:**

The paper proposes a unified transformer based model for Image-Language and Video-Language tasks by performing both image-language and video-language pretraining. The authors propose a decoupled joint pretraining image-language and video-language in order to obtain a boost in performance on multiple downstream tasks. The model is pretrained in a decoupled way, where first it is pretrained on image-language and then on video-language different from prior works. Moreover, the authors propose a new loss function that leverages image-text, video-text, image-label and video-label information. Finally, the authors test their proposed approach on multiple benchmarks.

**Questions:**

1. Can you give more details on the data used for the experiments from Table 9? How many samples are there? Do you use the "label" anotation?

2. Does the Table 9 experiments assume a zero-shot setup, or the model is fine-tuned on the downstream task? If it is finetuned, can you also provide numbers without pre-training?

3. I find a bit strange the fact that "Joint" pre-training performs worse than "Image-only" and also that "img2vid" brings an improvement while "joint" doesn't. Do you have any insights about this? Also, I would suggest including some of the "img2vid" results in the main paper.

4. The decoupled learning seems to bring a quite significant gain. I would emphasize this aspect a bit more.

5. Can you give more details about the "*" from the tables. From my understanding 14M* means that the model was pre-trained on 14M text-image pairs as well as on text-video data. Is this correct? If yes, please make it more obvious in the paper. Also, if this is indeed the case, I think it would be fair to include the numbers only with the 14M samples to have a clear comparison on the amount of performance the video data brings.

**Limitations:**

The limitations are briefly discussed while the potential societal impact is left for future studies.

**Strengths And Weaknesses:**

Strengths
The paper tackles an important problem and proposes an interesting system that achieves good results. The overall paper is fairly well written and the overall message is well understood.

Weaknesses
While the overall idea is interesting and the results look promising, I feel like the ablation part of the paper could be more comprehensive. So, I have several questions related to these ablations (please see below)

Missing citations
* Miech, Antoine, et al. "Thinking fast and slow: Efficient text-to-visual retrieval with transformers." Proceedings of the IEEE/CVF Conference on Computer Vision and Pattern Recognition. 2021.
* Croitoru, Ioana, et al. "Teachtext: Crossmodal generalized distillation for text-video retrieval." Proceedings of the IEEE/CVF International Conference on Computer Vision. 2021.
* Bogolin, Simion-Vlad, et al. "Cross Modal Retrieval with Querybank Normalisation." Proceedings of the IEEE/CVF Conference on Computer Vision and Pattern Recognition. 2022.

---

> ### Author Response · Authors · 2022-08-02
> **Thanks for your valuable comments**
>
> Thank you very much for recognizing the importance of our work. We are encouraged that the reviewer think our paper is well-written and our results are very promising. We also thank the reviewer for the valuable comments about the ablation part and have polished our paper based on the suggestions. Below are the detailed response.
>
> **Q1: Missing references.**
>
> Answer: Thanks for pointing this out, we have cited and discussed these papers in the revision.
>
> **Q2: More details for the data used in Table 9.**
>
> Answer: Thanks for raising this question! For a fair comparison, we use both visual-label data and vision-language data here. The detailed settings are listed below.
>
> | Pretraining | # Image-Text | # Image-Label | # Video-Text | # Video-Label |
> |:---|:---:|:---:|:---:|:---:|
> | Video-only | | | 2.5M | 0.3M |
> | Image-only | 14M | 1.3M | | |
> | Joint | 14M | 1.3M | 2.5M | 0.3M |
> | Img2vid | 14M | 1.3M | 2.5M | 0.3M |
> | Decoupled Joint | 14M | 1.3M | 2.5M | 0.3M |
>
> ***We have added these details in the supplementary material and the corresponding description in the main paper***.
>
> **Q3: Does the Table 9 experiments assume a zero-shot setup, or the model is fine-tuned on the downstream task? If it is finetuned, can you also provide numbers without pre-training?**
>
> Answer: Thanks for the great question! The results in Table 9 are evaluated under a fine-tuning setting. We provide the numbers without pretraining below. By comparison, we can see that pretraining on large-scale data could improve the results on all types of downstream tasks, and our proposed decoupled joint pretraining brings the most significant performance improvement. ***We have integrated these results into the Table 9 of the revision and added corresponding text description.***
>
> | Pretraining | COCO TR@1 | COCO IR@1 | MSRVTT (ret) | COCO (cap) B@4 | COCO (cap) C | VQA | MSRVTT(QA) |
> |:---|:---:|:---:|:---:|:---:|:---:|:---:|:---:|
> | Without Pretraining | 37.1 | 28.5 | 9.6 | 27.4 | 80.0 | 39.51 | 36.6 |
> | Decoupled Joint | 82.1 | 64.8 | 47.8 | 39.8 | 133.9 | 78.33 | 44.1 |
>
> **Q4: I find a bit strange the fact that "Joint" pre-training performs worse than "Image-only" and also that "img2vid" brings an improvement while "joint" doesn't. Do you have any insights about this? Also, I would suggest including some of the "img2vid" results in the main paper.**
>
> Answer: Really great question! The possible reason is that, as video data contain complex spatial-temporal information, it is difficult (or needs very long pretraining epochs) for the model to converge by performing video pretraining (or joint pretraining) from scratch. This is partially reflected by the bad results of "Video-only" pretraining. By contrast,  "img2vid" first performs image pretraining before conducting the video pretraining, which provides a strong initialization by first learning the spatial representation well and thus makes the model easier to learn in the following video pretraining. ***We have followed your suggestion and moved the "img2vid" results into the main paper***.
>
> **Q5: The decoupled learning seems to bring a quite significant gain. I would emphasize this aspect a bit more.**
>
> Answer: Thanks for your great suggestion! ***We have added more discussion and emphasis in the revision***.
>
> **Q6: Can you give more details about the "$\*$" (or 14M$\^\{\*\}$ ) from the tables. From my understanding 14M means that the model was pre-trained on 14M text-image pairs as well as on text-video data. Is this correct? If yes, please make it more obvious in the paper. Also, if this is indeed the case, I think it would be fair to include the numbers only with the 14M samples to have a clear comparison on the amount of performance the video data brings.**
>
> Answer: Yes, your understanding is correct! **14M$\^\{\*\}$** denotes extra video data is used (as mentioned in line 217, we have highlighted it in the revision Line 210). Actually, we have shown the results with 14M image only data in Table 9 (denoted as **Image-only**) and supplementary material.

---

> > ### Comment · Reviewer_VLaG · 2022-08-07
> > **Thank you for the response**
> >
> > Hi,
> >
> > The provided response answer my questions. Thanks!

---

> > > ### Author Response · Authors · 2022-08-07
> > > **Thanks for confirming!**
> > >
> > > Hi reviewer VLaG, thanks for your confirming! Really grateful!

---

> ### Author Response · Authors · 2022-08-04
> **Help check the rebuttal and happy to discuss more**
>
> Hi Reviewer VLaG,
>
> We are very grateful for your efforts and positive feedbacks. Can you help check our response and see whether your questions are well answered? We are happy to discuss with you about any remaining questions you may still have.

---

### Official Review · Reviewer_cPTx · 2022-07-11

**Rating:** 5
**Confidence:** 5
**Soundness:** 2 fair
**Presentation:** 2 fair
**Contribution:** 2 fair

**Summary:**

The authors propose a new foundation model for image-language and video-language tasks.
First, it applies a single unified transformer-based visual encoder for both image and video inputs. Second, they propose a decoupled joint pre-training of image-language and video-language to effectively leverage spatial and temporal dimensions.
Third, they propose a unified vision-language contrastive loss to leverage different sources of data.
The authors argue they achieve a good performance on many different types of tasks.


**Questions:**

	1. The performance comparison is confusing. Why do you report the performance of SimVLM in table7 but do not report it in table6? As we know, SimVLM is able to do both VQA and captioning. This is common for other baselines in other tables.
	2. In Section2, the authors claim FLAVA is concurrent work, which is not true. FLAVA is proposed last year and their results should be included and discussed.



**Limitations:**

1. include more studies and conduct a fair and comprehensive comparison

**Strengths And Weaknesses:**

Strengths:
	1. The motivation is good. The authors aim to build a foundation model for image-language and video-language tasks, which is a popular research topic.


Weaknesses:
	1. The so-called decoupled paradigm separates the image-language pretraining and video-language joint pretraining. The two-stage pre-training is a bit complicated. End-to-end pre-training is easier to train.
	2. Several important baselines are omitted. For example, FLAVA serves as an important foundation model but there are few comparisons. Why? I found that FLAVA outperforms Omnivl on ood101, DTD, and Flowers. Also, for image captioning tasks, the state-of-the-art is already 140+. Although different model sizes and pre-training data are used, the authors should compare with more recent methods (like SimVLM [1], OFA [2]).
	3. Several important datasets are omitted. For example, Imagenet is an important dataset for linear probing but this paper did not report its performance.
	4. Because a lot of important baseline models and datasets are missing, the comparison is not comprehensive.

[1] SimVLM: Simple Visual Language Model Pretraining with Weak Supervision
[2] OFA: Unifying Architectures, Tasks, and Modalities Through a Simple Sequence-to-Sequence Learning Framework

---

> ### Author Response · Authors · 2022-08-02
> **Thanks for your valuable comments**
>
> **Q3: Also, for image captioning tasks, the state-of-the-art is already 140+. Although different model sizes and pre-training data are used, the authors should compare with more recent methods (like SimVLM, OFA).**
>
> Answer: We don’t include SIMVLM and OFA for comparisons on image caption because the pretraining data that they adopt are far larger than ours (1.8 B / 21.4 M v.s. 14 M). Even so, OmniVL still achieves comparable or even better results on some metrics, e.g., SPICE, on the COCO Caption dataset.
>
> For fair comparisons, we pretrain OFA on the 14M image-text data with the officially released codebase (with only image-text matching and image captioning objectives, denoted as OFA$\_\{14M\}$), the results (row 2 in the table below) demonstrate that using the same amount of pre-training data, OmniVL performs better than OFA measured by all the metrics. Note that we don't show the results of SIMVLM on 14M data since they haven't released their code.
>
> | Method | # Pre Img-Txt | B@4 | METEOR | CIDER | SPICE |
> |:---|:---:|:---:|:---:|:---:|:---:|
> | SIMVLM | 1.8B | 39.0 | 32.9 | 134.8 | 24.0 |
> | OFA$\_\{14M\}$ | 14M | 38.7 | 30.6 | 130.5 | 23.5 |
> | OFA | 21.4M | 41.0 | 30.9 | 138.2 | 24.2 |
> | OmniVL | 14M | 39.8 | 31.2 | 133.9 | 24.2 |
>
> Additionally, with less pretraining data, OmniVL outperforms SIMVLM and OFA on visual question answering, which also validates the effectiveness of our method.
>
> | Method | # Pre Img-Txt | test-dev | test-std |
> |:---|:---:|:---:|:---:|
> | SIMVLM | 1.8B | 77.9 | 78.1 |
> | OFA | 21.4M | 78.0 | 78.1 |
> | OmniVL | 14M | 78.3 | 78.4 |
>
> We have included all the new comparison results in the uploaded revision.
>
> **Q4: Imagenet ommited for linear probing.**
>
> Answer: Great question! We omitted ImageNet for linear probing evaluation because it is included in our UniVLC pretraining data, in that case, it is unfair to compare the linear probing performance of OmniVL with other methods.
>
> **Q5: Incorrect description about FLAVA in Section2.**
>
> Answer: Thanks for pointing it out! We have fixed this issue in the uploaded revision.

---

> > ### Author Response · Authors · 2022-08-03
> > **Is there any more question or concern?**
> >
> > Hi Reviewer cPTx, thanks for your effort and suggestions! We have addressed the concerns in the above rebuttal, can you help take a look and see whether your concerns are well addressed? We are happy to answer any question you may still have.

---

> > > ### Author Response · Authors · 2022-08-07
> > > **Any more question or concern?**
> > >
> > > Hi Reviewer cPTx, since the reviewer-authors discussion time window will be closed soon, can you help read the rebuttal as soon as possible so that we can address any remaining concerns you may have. Grateful for your effort!

---

> > > > ### Comment · Reviewer_cPTx · 2022-08-08
> > > > **Answer to rebuttal**
> > > >
> > > > Dear authors,
> > > > Thanks for your response.
> > > > My concerns with Q1, Q4, Q5 are addressed. However, for Q2 and Q3, I still have doubts about the comparison.
> > > >
> > > > **[Comparisions are not fair]**
> > > >
> > > > I agree that some comparisons are not fair because those methods are trained on different datasets (e.g., SimVLM on 1.8B) but the comparisons with OFA and FLAVA are not convincing.
> > > > For example, pre-training OFA on only image-text matching and image captioning objectives is not a correct way to use OFA because it is a multi-tasking model. Removing other objectives would definitely compromise its performance.
> > > > So here is another question, is this approach scalable to large pre-training data? If so, could you report the performance pre-training with FLAVA's data or OFA's data?
> > > >
> > > > **[NoCaps experiments are invalid]**
> > > >
> > > > In addition, as for the NoCaps experiments, I do not think it is the correct way to do NoCaps experiments.
> > > > If you are familiar with this dataset, you will know the initial proposal of NoCaps does not allow the introduction of extra image-text pairs [1]. Most methods (e.g., VinVL) followed this rule. However, this work adopted the pre-trained and then fine-tuned model on COCO to evaluate on NoCaps. So the experiments on NoCaps are invalid.
> > > >
> > > > Anyway, I appreciate the authors' efforts in the additional experiments.
> > > > I am glad to raise my rating but I still lean to reject.
> > > >
> > > >
> > > > [1] Agrawal, H., Desai, K., Wang, Y., Chen, X., Jain, R., Johnson, M., ... & Anderson, P. (2019). Nocaps: Novel object captioning at scale. In Proceedings of the IEEE/CVF International Conference on Computer Vision (pp. 8948-8957).

---

> > > > > ### Author Response · Authors · 2022-08-08
> > > > > **Thanks for your reply**
> > > > >
> > > > > Before we answer your questions, may we ask two quick questions?
> > > > >
> > > > > 1. We have reported the results of FLAVA on 14M image-text data, could you please provide some details on why you think the comparison with FLAVA is not convincing enough?
> > > > >
> > > > > 2. Does "extra image-text pairs" in your second question refer to Image-text pairs adopted in pretraining?

---

> > > > > ### Author Response · Authors · 2022-08-08
> > > > > **Answers for your new questions-Part1**
> > > > >
> > > > > Dear reviewer cPTx,
> > > > >
> > > > > Considering the reviewer-author discussion time window will be closed soon, we first answer your questions based on our understanding. Feel free to raise any more questions if our answers do not exactly fit your questions.
> > > > >
> > > > > **Q1: Comparisons are not fair/convincing?**
> > > > >
> > > > > Answer: To answer this question more clearly, we split it into three sub-questions:
> > > > >
> > > > > 1) For the comparison with FLAVA, our comparison is **totally fair and convincing**. Even though FLAVA is pretrained on a larger data than ours, our OmniVL outperforms FLAVA on most tasks/datasets. For example, on image-text retrieval task, our OmniVL achieves better results than FLAVA on both COCO and Flickr under zero-shot and fine-tuning settings (see answer to your question 2 in the rebuttal above). For linear probing results (please see all results in the supplementary material), even though FLAVA performs better than our OmniVL on Food101, DTD and Flowers datasets, our OmniVL outperforms FLAVA on CIFAR10, CIFAR100 and Pets datasets. And the average performance on these six datasets of our OmniVL is **even slightly better** than FLAVA(**86.7 vs 86.3**). In this sense, even under this unfair setting (FLAVA uses larger dataset than ours), our OmniVL is already better than FLAVA. By pretraining FLAVA on the same 14M image-text data (fair setting, FLAVA$\_\{14M\}$), we demonstrate our OmniVL outperforms FLAVA on most datasets. Detailed table is below.
> > > > >
> > > > > | Method | Food101 | CIFAR10 | CIFAR100 | Pets | DTD |  Flowers | Avg |
> > > > > |:---|:---:|:---:|:---:|:---:| :---:| :---:| :---:|
> > > > > | FLAVA    | 85.2  | 90.4 | 76.2 | 82.3 | 74.2 | 92.7 | 83.5 |
> > > > > | FLAVA$\_\{14M\}$ | **88.5** | 92.9 | 77.7 | 84.8 | **77.3** | **96.4** | 86.3 |
> > > > > | OmniVL | 87.4 | **96.2** | **83.2** | **87.1** | 76.5 | 89.8 | **86.7** |
> > > > >
> > > > > 2) For the comparison with OFA, as you said, our OmniVL and OFA **target different research perspective, therefore it is indeed hard or even impossible to compare them under an exactly fair setting**. In details, OFA targets multi-task unification in the image/text domain, including detection, captioning, VQA, Image Infilling, visual grounding, and so on. On contrary, our OmniVL mainly proposes to unify image-language and video-language pretraining. Technically, they are orthogonal efforts that study different things. If we  involve other tasks beyond image-text matching and  image captioning, then the comparison is unfair for us because our OmniVL does not use the pretrainging data of such tasks. To summarize, only involving image-text matching and image captioning may not look like a totally fair/convincing setting, but we have tried our best to do that. Based on the fact that OFA and our OmniVL are orthogonal effort, we would like to ask you to understand our situation.
> > > > >
> > > > > 3) For the scalability of large pre-training data, **we have already conducted the corresponding experiments in Table 4 of our main paper**. By scaling the pretraining image-text pair data from 4M to 14M, we have observed significant performance gain, which well demonstrates the scalability of our method. We understand pretraining on a even larger scale like FLAVA, OFA, or even SIMVLM may make the results more solid, but our GPU resources cannot support to finish such experiments in a short time (e.g., the estimated training time on FLAVA 70M data will take more than three weeks). Considering most research teams like us do not have many gpus like big companies like google (SIMVLM), facebook (FLAVA) and Alibaba (OFA), we hope you can understand it. But we may emphasize that **14M and 4M image-text data settings are the most common setting for the research community in this direction**. For example, baseline methods including UNITER, OSCAR, UNIMO, VLMO, ALBEF and BLIP all follow such settings. More importantly, **the performance gain by scaling image-text data from 4M to 14M can well demonstrate the scalability of our OmniVL**. But anyway, if pretraining on even larger data is the only factor that obstacle you from recommending acceptance, we promise to add such results into the camera ready version (more time left for us).

---

> > > > > ### Author Response · Authors · 2022-08-08
> > > > > **Answer for your new questions-Part2**
> > > > >
> > > > > **Q2: NoCaps experiments are invalid?**
> > > > >
> > > > > Answer: Great question! We agree that "the initial proposal of NoCaps may not allow the introduction of extra image-text pairs" when it was first proposed in year 2018, however, Vision-Language Pertraining (VLP) has been rapidly developing in recent years and it is a common practice to evaluate the pretrain-then-finetune model on NoCaps validation for VLP methods now. **All baselines [1][2][3][4][5] except OFA (results not reported) in Table 6 follow the same setting as our OmniVL**, i.e., use image-text pairs during pretraining to evaluate the pretraining performance. Especially ***note that although VinVL reports the results without pre-training, the numbers in our paper are reproduced by LEMON[3] via finetuning from the released checkpoints (which is pre-trained on the combined datasets including 5.65M images, 2.5M QAs, 4.68M captions and 1.67M pseudo-captions)***. In this sense, our comparison is at least fair.
> > > > >
> > > > >
> > > > > [1] Conceptual 12m: Pushing web-scale image-text pre-training to recognize long-tail visual concepts.
> > > > >
> > > > > [2] VinVL: Revisiting Visual Representations in Vision-Language Models.
> > > > >
> > > > > [3] Scaling up vision-language pre-training for image captioning.
> > > > >
> > > > > [4] Blip: Bootstrapping language-image pre-training for unified vision-language understanding and generation.
> > > > >
> > > > > [5] SimVLM: Simple Visual Language Model Pretraining with Weak Supervision.
> > > > >
> > > > > Feel free to discuss with us about any new concerns that may obstacle you from changing your decision into accept! We are grateful for your efforts!
> > > > >
> > > > > We notice you gave relatively low scores in terms of  **Soundness**,  **Presentation**, and **Contribution**, do you have specific concerns in terms of these aspects. Also feel free to raise any suggestions for them, we are happy to polish our paper with another revision or in the camera ready version.

---

> > > > > ### Author Response · Authors · 2022-08-09
> > > > > **Help read the response to your new questions.**
> > > > >
> > > > > Hi cPTx, sorry for disturbing you again and really thanks for your effort! Considering the reviewer-author discussion time window will be closed in 15 hours, can you help read our response to your new questions at your earliest convenience? We hope our response can address your only last two concerns and are also happy to discuss with you further.

---

> > > > > ### Comment · Reviewer_cPTx · 2022-08-09
> > > > > **Ansewr to rebuttal**
> > > > >
> > > > > Thanks for your rebuttal, I appreciated the newly added experiments.
> > > > > The comparison becomes stronger now. My concerns about NoCaps are addressed.
> > > > > I have updated my ratings.

---

> > > > > > ### Author Response · Authors · 2022-08-09
> > > > > > **Thanks for your effort**
> > > > > >
> > > > > > Dear Reviewer cPTx, thanks for your effort again! We are happy that our rebuttal well addressed your concerns!

---

> ### Author Response · Authors · 2022-08-02
> **Thanks for your valuable comments**
>
> Thank you for your valuable feedback. **We have addressed all the concerns and added missing comparisons in the revision**. We hope it can convince you and change the decision.
>
> **Q1: The so-called decoupled paradigm separates the image-language pretraining and video-language joint pretraining. The two-stage pre-training is a bit complicated. End-to-end pre-training is easier to train.**
>
> Answer: Great question! We know one-stage end-to-end pre-training may look easier to train, but the complex spatial-temporal information within videos indeed makes it difficult to learn video and video-language representation from scratch. It is not only inefficient but also ineffective. The ineffectiveness is reflected by the results in Table 9, where the proposed decoupled joint pretraining outperforms both ***Joint*** Pretraining (which is trained end-to-end) and ***Video-only*** Pretraining (which is trained end-to-end) by a large margin. The proposed decoupled joint pretraining is also the **key** to make image-language and video-language benefit each other. The inefficiency can be partially reflected by the recent work TimeSformer[1] and video self-supervised pretraining work VideoMAE [2], which requires more than 800 training epochs.
>
> We would also like to point out that, stage-wise pretraining has been widely adopted by other VL pretraining methods, e.g., FLAVA, VLMO, and e.t.c, due to its simplicity in implementations as well as its effectiveness.
>
> [1] Is Space-Time Attention All You Need for Video Understanding?
>
> [2] Masked Autoencoders As Spatio-temporal Learners.
>
> **Q2: Several important baselines are omitted. FLAVA serves as an important foundation model but there are few comparisons. Why? I found that FLAVA outperforms Omnivl on Food101, DTD, and Flowers.**
>
> Answer: Since FLAVA only reports the zero-shot image-text retrieval results in their paper, and it doesn't support image captioning (lacking a text decoder for text generation), we don’t compare with it for these two tasks.
>
> To make a comparison with FLAVA, we evaluate the fine-tuning results of FLAVA on our own, and compare the image-text retrieval performance of FLAVA and OmniVL on both COCO (top) and Flickr (down) under zero-shot and fine-tuning settings. The results below demonstrate OmniVL outperforms FLAVA by large margins on both datasets.
>
> | Method | FT (TR@1/5/10) | FT (IR@1/5/10) | ZS (TR@1/5/10) | ZS (TR@1/5/10) |
> |:---|:---:|:---:|:---:|:---:|
> | FLAVA | 61.5 / 82.1 / 89.6 | 50.1 / 74.4 / 83.2  | 42.7 / 76.8 / - | 38.4 / 67.5 / - |
> | OmniVL | 82.1 / 95.9 / 98.1 | 64.8 / 86.1 / 91.6 | 71.8 / 90.6 / 95.0 | 56.4 / 80.8 / 87.8 |
>
>
> | Method | FT (TR@1/5/10) | FT (IR@1/5/10) | ZS (TR@1/5/10) | ZS (TR@1/5/10) |
> |:---|:---:|:---:|:---:|:---:|
> | FLAVA | 85.4 / 95.7 / 98.3 | 73.2 / 92.7 / 95.5 | 67.7 / 94.0 / - | 65.2 / 89.4 / - |
> | OmniVL | 97.1 / 99.8 / 100.0 | 87.5 / 97.5 / 99.0 | 87.8 / 98.3 / 99.2 | 76.1 / 92.5 / 95.4 |
>
> Although FLAVA outperforms OmniVL on Food101, DTD, and Flowers in terms of linear probing, we argue that their pretraining data (70M image-text data) are much larger than ours. For fair comparisons, we pretrain FLAVA (we adopt the implementation in [torchmultimodal](https://github.com/facebookresearch/multimodal/tree/main/examples/flava)) on the 14M image-text data (denoted as FLAVA$\_\{14M\}$) and evaluate the results using linear probing on Food101, CIFAR10, CIFAR100, Pets, DTD, and Flowers. The results below demonstrate that using the same amount of pretraining data, OmniVL beats FLAVA on most datasets. (Due to space limit, we put the linear probing comparison results with FLAVA in the supplementary material but will move to the main paper in the camera ready version, which has an additional content page space.)
>
> We also compare OmniVL with FLAVA$\_\{14M\}$ on COCO retrieval through finetuning. We see from the last two columns below that OmniVL outperforms FLAVA$\_\{14M\}$ clearly.
>
>
> | Method | Food101 | CIFAR10 | CIFAR100 | Pets | DTD | Flowers | COCO FT TR@1 | COCO FT IR@1 |
> |:---|:---:|:---:|:---:|:---:|:---:|:---:|:---:|:---:|
> | FLAVA$\_\{14M\}$ | 85.2 | 90.4 | 76.2 | 82.3 | 74.2 | 92.7 | 57.4 | 48.7 |
> | OmniVL | 87.4 | 96.2 | 83.2 | 87.1 | 76.5 |  89.8 | 82.1 | 64.8 |

---

### Official Review · Reviewer_5rq1 · 2022-07-13

**Rating:** 7
**Confidence:** 3
**Soundness:** 4 excellent
**Presentation:** 3 good
**Contribution:** 3 good

**Summary:**

This paper proposes OmniVL, a vision-language foundation model that supports both image-language and video-language pretraining in a unified framework. The model allows for evaluation of vision-only tasks, multimodal retrieval, and multimodal generation based tasks like captioning and VQA. The model achieves comparable or SOTA performance across many benchmark datasets.

**Questions:**

- Why does OmniVL perform more poorly on the in-domain subset of the NoCaps captioning task?

**Limitations:**

I don't think the authors adequately addressed limitations of the approach or negative societal impact. Computational resources were broadly mentioned, but the societal consequences sentences are hand wavy and it gives the impression the authors didn't think critically about this.

**Strengths And Weaknesses:**

Strengths
- The proposed framework unifies different modality objectives, downstream task formulations, and input data types.
- Given the current move toward "foundation" models, this work seems like an obvious next step toward general multimodal models that are performant in a variety of downstream tasks, and is new to incorporate video data at the same time.
- There are comprehensive experiments and performance is either comparable or SOTA on some downstream evaluation tasks.
- The decoupled pretraining is a sensible idea and has a large impact on performance given the ablation study

Weaknesses
- While the paper states that their unified vision-language contrastive (UniVLC) is a novel contribution, it is a straight forward extension of the existing UniCL loss.

---

> ### Author Response · Authors · 2022-08-02
> **Thanks for your valuable comments**
>
> Thank you for acknowledging the importance of our work and sharing the valuable feedback! Below, we answer the questions one by one.
>
> **Q1: More discussion about limitation and negative societal impact.**
>
> Answer: Thanks for your suggestion! We have added more discussion about limitation and negative societal impact in the revision. Here are updated section: ***Although our model has achieved superior results on a wide range of downstream tasks, it still lacks of the commonsense reasoning capability required by some visual-language interaction tasks (e.g., visual/video question answering). It also needs better architecture design to enable the zero-shot capability for visual question answering and few-shot task customization capability like GPT-3. From the societal impact perspective, since our model is pretrained on the large-scale web-crawled data which may contain some toxic language or bias, and it is not easy to explicitly control the model output, much attention should be paid to ensure responsible model deployment.***
>
> **Q2: Why does OmniVL perform more poorly on the in-domain subset of the NoCaps captioning task?**
>
> Answer: As mentioned in Line 259, we adopted the fine-tuned model on COCO to evaluate on NoCaps, and the results in Table 6 demonstrate that OmniVL achieves better performance on the overall dataset. Considering the in-domain subset of NoCaps share a similar set of object classes with COCO, we think the slightly worse results (15 v.s. 15.1 in terms of SPICE) compared with BLIP might possibly result from the fact that OmniVL doesn’t overfit on COCO, producing decent results on both COCO and the in-domain split of NoCaps. More importantly, the resulting model does generalize to new domains---we observe significant performance gains produced by OmniVL compared to alternative methods on near-/out-of domain splits. This suggests that OmniVL is able to balance the tradeoff between overfitting and generalization. This is particularly encouraging given that generalization ability is arguably the gold standards to evaluate the performance of machine learning models.
>
>
> **Q3: UniVLC is a straightforward extension of UniCL loss.**
>
> Answer: Thanks. In this paper, we aim to achieve unification in three dimensions: modality, functionality, and pretraining corpus. To this end, we build upon UniCL and make the following changes to fully utilize both webly-crawled visual-text data and human-annotated visual-label data. First, we maintain memory banks to store the most recent visual vectors and text vectors from momentum encoders, which facilitates us to enjoy a large batch size for contrastive learning. Second, we further extend its scope to cover video data. We agree that this extension is very intuitive and straightforward, but we are the first that demonstrates its effectiveness in unifying image-language and video-language pretraining.

---

> > ### Comment · Reviewer_5rq1 · 2022-08-07
> > **Response to comments**
> >
> > Hi Authors,
> >
> > Thanks for addressing my comments and for your patience in my response. I am fine with your response to Q1 and Q3, but have some follow up questions on Q2.
> >
> > 1. Yes, the spice compared with BLIP is 15 vs. 15.1. However, there is a larger gap on CIDEr: 104.6 vs. 111.3.
> > 2. I actually was referring to why does OmniVL do better on near and out of domain splits than it does on in domain samples, since those are typically the highest performing split (regardless of the other models you compare to). Is your explanation for this because it's only fine-tuned on COCO?

---

> > > ### Author Response · Authors · 2022-08-08
> > > **Thanks for your reply and further questions**
> > >
> > > Hi Reviewer 5rq1, thanks for your response! You really asked a very good & interesting question! To answer this question in detail, we would like to first discuss the two metrics "CIDEr" and "SPICE". "CIDEr" is a n-gram based metric, which measures the word overlap between generated and reference captions. That is to say, it will be higher if the predicted captions and reference captions have the exactly same words. This property also makes it sensitive to n-gram overlap, thus cannot well evaluate cases where two sentences are semantically similar but with different words. On the contrary, the "SPICE" metric[1] measures the semantical similarity of scene graphs constructed from predicted and reference caption but not n-gram, therefore it shows better correlation with human judgement (More detailed can be found in [1]). Therefore, the close performance in terms of SPICE indicates our model can generate **semantically similar results** as BLIP.
> > >
> > > Answer for sub-question 1: Based on the above analysis, the large performance gap in terms of the CIDEr metric and similar SPICE performance indicates that our model generates semantically similar captions but with different words. The possible reason may be that we conduct pretraining on more diverse datasets, i.e., involving both image-language data and video-language data, rather than only image-language data as BLIP. Since we cannot see the groundtruth captions during evaluation (submitted to the server), we are unable to prove our hypothesis by checking detailed examples.
> > >
> > > Answer for sub-question 2: To be honest, this question is very interesting but difficult to answer. Indeed, we also observe similar phenomenon in recent SOTA methods in the NoCaps official leaderboard, i.e., achieving better results (in terms of CIDEr) on near and out of domain splits than in-domain split on the test set. For example, the CIDEr of Rank-1st method GiT on in-domain and near-domain are 122.40, 123.92 respectively, the CIDEr of Rank-2nd method CoCa on in-domain, near-domain and out-domain are 117.90, 120.73 and 121.69 respectively. We also discussed this phenomenon with experts of image and video captioning tasks, but cannot find very concrete explanations because the groundtruth captions of the test set are not accessible. We guess the possible reason may come from two aspects: 1) There is a tradeoff between overfitting and generalization, i.e., good generalization performance on near-domain and out-domain will lead to slightly worse results on in-domain split. 2) The three splits of the test set may have different annotation distribution.
> > >
> > >
> > > [1] Spice: Semantic propositional image caption evaluation

---

> > > > ### Author Response · Authors · 2022-08-09
> > > > **Any further question?**
> > > >
> > > > Hi Reviewer 5rq1, thanks again for your effort. Do you have any more concerns?  As the reviewer-authors discussion window will be closed in about 18 hours, please raise your questions as soon as possible if existed. We are happy to answer them! Grateful for your help!

---

> ### Author Response · Authors · 2022-08-04
> **Help check whether questions are well answered.**
>
> Dear reviewer 5rq1,
>
> We would like to thank you again for your effort and positive feedback. Can you help find time to take a look at the response and check whether your questions are well answered. We are very happy to discuss with you and provide further clarification for any new question.

---

### Author Response · Authors · 2022-08-02
**We thank all reviewers for their valuable comments**

We thank all reviewers for their valuable comments. We are happy all reviewers think our paper addressed an important research topic with good motivation. We are also encouraged that Reviewer 5rq1 think our work is an obvious next step toward general multimodal models, and Reviewer VLaG thinks our paper is well-written and overall message is well understood.

**We have revised our paper and updated it in the system**. In the revision, 1) We have followed the suggestion of Reviewer 5rq1 and added more in-depth discussion of limitation and negative societal impact; 2) We have addressed the main concerns of Reviewer cPTx  by adding comparisons with more SOTA methods include FLAVA, SIMVLM and OFA, and fixed the imprecise description in Section 2; 3) Based on the suggestion of Reviewer VLaG, we have added the missing references and the ablation result without pre-training in Table 9,  moved the "img2vid" results into main paper, and emphasized decoupled learning more.

---

### Meta-Review · Area_Chair_K7WV · 2022-08-24

**Recommendation:** Accept
**Confidence:** Certain

**Metareview:**

After the authors’ rebuttal and long discussion between reviewers and authors, the paper unanimously receives positive rates thanks to reasonable proposed ideas and thorough experiment evaluation. The camera-ready version may need to be updated to fully reflect reviewers’ comments and authors’ answers to them.

**Award:**

No

---

### Decision · Program_Chairs · 2022-09-14

Accept